# The Effects of Combined 1-Methylcyclopropene and Melatonin Treatment on the Quality Characteristics and Active Oxygen Metabolism of Mango Fruit during Storage

**DOI:** 10.3390/foods12101979

**Published:** 2023-05-12

**Authors:** Fang Yuan, Chunyan Wang, Ping Yi, Li Li, Guifen Wu, Fang Huang, Min Huang, Ting Gan

**Affiliations:** 1College of Chemistry and Biological Engineering, Guangxi Minzu Normal University, Chongzuo 532200, China; yf2018yuyu@126.com (F.Y.);; 2Agro-Food Science and Technology Research Institute, Guangxi Academy of Agricultural Sciences, Nanning 530007, China; pingyi@gxaas.net (P.Y.);; 3Guangxi Key Laboratory of Fruits and Vegetables Storage-Processing Technology, Guangxi Academy of Agricultural Sciences, Nanning 530007, China

**Keywords:** 1-methylcyclopropene, melatonin, mango fruit, active oxygen metabolism, storage

## Abstract

In this study, mango fruit (Tainong No. 1) was treated with either 0.1 mg/L 1-methylcyclopropene (1-MCP) alone or with a combination of 0.1 mg/L 1-MCP and 0.2 mM melatonin (MT). The mango fruit was then stored for 10 days at 25 °C and 85–90% relative humidity. Quality characteristics and the active oxygen metabolism of postharvest mangoes were evaluated every 2 days. Compared to untreated mango fruit, those with the treatments of 1-MCP alone or 1-MCP + MT had a better appearance and higher levels of soluble sugar, ascorbic acid, and titratable acidity. Moreover, these treatments prevented the loss of fruit firmness, successfully delayed the escalation of *a** and *b** values, and reduced malondialdehyde content and superoxide anion generation rate. After 10 days of storage, mango fruit treated by 1-MCP alone or 1-MCP + MT exhibited increased activities of antioxidant enzymes such as ascorbate peroxidase, catalase, superoxide dismutase, and other peroxidases; nevertheless, the two treatment protocols maintained higher mango total phenolic content only at the later stage of storage. These findings suggest that mango fruit treated with 1-MCP alone or with 1-MCP + MT improves the quality characteristics and antioxidant activities. Moreover, compared to 1-MCP treatment alone, 1-MCP + MT-treated mangoes exhibited higher quality and a stronger regulation of active metabolism during storage.

## 1. Introduction

Mango (*Mangnifera indica* L.), the ‘King of Fruits’, is grown worldwide for its soft and juicy pulp, unique flavour and high nutritional content [1,2]. Mangoes can provide plenty of antioxidants for the human body such as phenolics and vitamins [3,4]. However, mango fruit is not postharvest-storage resistant because of its vigorous metabolism. It matures and softens quickly at room temperature and then decays [5], reducing its nutritional and commercial values.

Previous studies have indicated that active oxygen metabolism has a significant impact on fruit ripening, senescence, and softening [6,7]. For instance, the single or combined use of tea polyphenolics and 1-methylcyclopropene (1-MCP) coating impeded the senescence of bracken and improved its storage quality by modifying its active oxygen metabolism and reducing its malondialdehyde content [6]. Exogenous H_2_O_2_ treatment has been found to accelerate the senescence and ripening processes of kiwifruit during late storage periods by leading to an imbalance in redox homeostasis [7]. Similar findings were obtained in studies on muskmelon [8], sweet cherry [9], strawberry [10], and mango fruit [11]. Although higher plants have antioxidant defence systems that protect tissues from excessive reactive oxygen species (ROS), external factors may also regulate reactive oxygen species metabolism.

1-MCP, a competitive inhibitor of ethylene, is safe and widely applied for preserving fruits and vegetables. Studies on pear [12], apple [13], nectarine [14], kiwifruit [15,16,17], and banana [18] have shown that 1-MCP can enhance the antioxidant activity, preserve the quality of postharvest fruit, reduce ROS production, and delay the senescence of fresh produce.

Melatonin (N-acetyl-5-methoxytryptamine, MT) is synthesized from serotonin, which is metabolized from tryptophan [19,20]. MT is commonly present in plants and animals and has direct or indirect regulatory influences on the metabolism and growth of plants [21,22,23,24]. Exogenous MT has excellent ROS scavenging ability, which has been shown to maintain higher activities of antioxidant enzymes (e.g., ascorbate peroxidase, catalase, glutathione reductase, peroxidase, and superoxide dismutase) [25,26], enhance the contents of antioxidants (e.g., glutathione, ascorbic acid, and phenolics [24]), and reduce the production of ROS [27]. Moreover, MT can maintain better fruit quality [25,26,27] and is related to ethylene synthesis in fruit [28].

The single use of MT or 1-MCP to promote the quality and shelf-life of mango has been investigated in many studies. However, studies on the influences of the combination of 1-MCP and exogenous MT on active oxygen metabolism and the nutritional quality of mango are limited. The optimum concentrations of MT and 1-MCP were selected in a pre-experiment, and based on this, the objective of this work was to assess the effect of the combined treatment of 1-MCP and MT on the firmness, nutrients, and antioxidant activity of fresh mango fruit during storage.

## 2. Materials and Methods

### 2.1. Fruit Treatments

Mangoes (Tainong No. 1) were collected at commercial maturity (pulp hardness, 151.70 N; total soluble solids content, 6.9%; titratable acid content, 10.21 mg/g Fw; and soluble sugar content 26.46 mg/g Fw). These were collected at 6:00 a.m. on 2 July 2022, from a well-managed orchard in Baise City, Guangxi Province, China. On the same day, mangoes were harvested and delivered to the laboratory. Mangoes were handpicked to remove those with visual defects, diseases, or mechanical damage. Mangoes were soaked for 10 min in a 30 mg/L chlorine dioxide solution for sterilisation and then washed with distilled water. About 25 washed fruits were randomly chosen for quality evaluation. The remaining mangoes were stochastically separated into three different groups (150 mangoes/group): 1-MCP, 1-MCP combined with MT (1-MCP + MT), and a control group (CK). Fruits were individually soaked for 30 min in the dark in distilled water (control), 1-MCP (0.1 mg/L), and a mixture of 1-MCP (0.1 mg/L) and MT (0.2 mmol/L); all treatments contained 0.01 mL/L Tween 20. After being immersed, the fruits were air-dried for 60 min at 25 °C in the dark. Subsequently, the treated fruits were exposed to 0 (control group) and 0.1 mg/L 1-MCP (1-MCP and 1-MCP + MT groups) for 12 h at 25 °C. Afterwards, the fruits were stored for 10 days at 25 °C and 85–90% relative humidity.

### 2.2. Determination of Colour Characteristics and Fruit Firmness

The colour characteristics of the surface of the mango fruits were evaluated using the approach reported by Chen et al. [29]. Three different individual mangoes were used for determination. The colour of two points on opposite sides of the equatorial perimeter of each mango was determined.

The mango fruit firmness was quantified based on the methods reported by Lin et al. [30] and Yu et al. [31], with minor modifications. A texture instrument (FTC/TMS-Pilot, Sterling, Virginia, USA) equipped with a 10 mm diameter probe (test rate, 60 mm/s, starting force, 0.38 N; and puncture distance, 10 mm) was employed for assessing the fruit firmness without the peel. Eight peeled fruits were analysed. The pulp on two opposite sides of the equatorial region was cut into squares (1.8 cm × 1.8 cm × 1.3 cm), which were used to determine the firmness.

### 2.3. Measurement of Total Soluble Solids, Soluble Sugar Content and Titratable Acidity

The total soluble solid content, soluble sugar content, and titratable acidity were calculated using the methods reported by Ma et al. [32], with minor modifications. The determination of total soluble solids in the filtrate was repeated five times. Titratable acidity was extracted from 8 g of pulp and determined by acid–base titration. The pulp from five mangoes was frozen in liquid nitrogen and 1.0 g of pulp powder was used to determine soluble sugar content. Total soluble solid values are expressed as percentages, whereas titratable acidity and soluble sugar concentrations are presented as mg/g on a fresh weight basis.

### 2.4. Quantification of Ascorbic Acid and Total Phenolic Content

The pulp from five mangoes was frozen in liquid nitrogen and crushed into a frozen powder, then the powdered pulp (3 g) was used to quantify the level of ascorbic acid using the approach of Lin et al. [33] and Chen et al. [34]. The ascorbic acid level is presented as mg/100 g on a fresh weight basis.

The total phenolic level was determined using the approach reported by Chen et al. [35], with slight modifications. Anhydrous methanol with 1% hydrochloric acid was used to extract total phenolic from 5 g of frozen mango pulp powder. The standard for total phenolic level determination was gallic acid (GA). The level of total phenolic is presented as mg GAE/kg (gallic acid equivalent).

### 2.5. Quantification of the Malondialdehyde Level and Superoxide Anion (O_2_^−.^) Generation Rate

The malondialdehyde level and O_2_^−.^ generation rate were quantified using the methods reported by Lin et al. [36,37], with minor modifications.

The pulp from five mangoes was frozen in liquid nitrogen and crushed into a frozen powder, and the powdered pulp (5 g) was then used to determine the malondialdehyde level and O_2_^−.^ generation rate. The level of malondialdehyde was calculated and shown as nmol/100 g on a fresh weight basis, and the O_2_^−.^ generation rate was displayed as μmol/min/kg on a fresh weight basis.

### 2.6. Assays of the Activities of Superoxide Dismutase, Catalase, Ascorbate Peroxidase, and Peroxidase

Superoxide dismutase, catalase, ascorbate peroxidase, and peroxidase activities were evaluated using an ELISA kit (Jiangsu Meimian Industrial Co., Ltd., Yancheng, China). The standard curve was plotted according to the standard’s activity and its corresponding OD value at 450 nm. The absorbance value of the enzyme activity in the sample was measured at 450 nm. The superoxide dismutase, catalase, ascorbate peroxidase, and peroxidase activities were reported as U/g on a fresh weight basis using the standard curve.

### 2.7. Statistical Analyses

All data are displayed as the average ± standard deviation. For statistical analysis, SPSS software (version 17.0) was employed to conduct a one-way analysis of variance. Different letters were applied to represent the statistical significance (*p* < 0.05).

## 3. Results

### 3.1. Colour Characteristics, Visual Appearance, and Fruit Firmness

The change in colour appearance is an important factor in determining fruit ripening and quality [29]. The noticeable change in mango after storage was that the peel turns yellow while the surface colour of the peel lightens (Figure 1E).

Within 4 days of storage, no significant change was observed in the *L** value of the mango epidermis (Figure 1A, Appendix A). However, after being stored for more than 4 days, the *L** value elevated. These results suggested that the colour of the mango epidermis became lighter (Figure 1E). Compared to the control mango, the *L** values increased more slowly in 1-MCP and 1-MCP + MT-treated mangoes. After a 6- to 10-day storage period, the control mango had a significantly higher *L** value than 1-MCP + MT-treated mango (*p* < 0.05). Furthermore, in the final 4 days of storage, the 1-MCP-treated mango also exhibited a higher *L** value than the 1-MCP + MT-treated mango.

The *a** value displayed an ascending trend with increased storage time (Figure 1B, Appendix A). The *a** values changed slowly in the 1-MCP + MT group. During storage days 0–6, no significant difference (*p* < 0.05) was reported in the *a** value between the 1-MCP and the control groups. Between 6 and 8 days of storage, *a** values in these two groups escalated rapidly, indicating that the mango peel’s greenish colour had reduced and quickly turned red during this time. A slow change in the *a** value was also observed in 1-MCP + MT-treated mango throughout the whole storage period. From the sixth day, the 1-MCP + MT group owned a significantly lower *a** value compared to the remaining two groups (*p* < 0.05).

During the 4-day storage, the *b** value changed slightly in all groups (Figure 1C, Appendix A) and then escalated speedily from the 6th day, indicating that the colour of the mango peel had begun to turn yellow (Figure 1E). The order of turning yellow from less to more was: 1-MCP + MT group, 1-MCP group, and then the control group (Figure 1E).

Firmness is one of the most important indicators of fruit quality [12,15]. As shown in Figure 1D and Appendix A, changes in mango firmness occurred along with the period of storage. The firmness of the mango decreased throughout the storage period, and rapid softening occurred within the storage duration (2–6 days). During the storage period, especially on the 4th day, the firmness degradation in mango fruits was significantly inhibited by 1-MCP + MT and 1-MCP. The firmness of mangoes treated with 1-MCP + MT and 1-MCP was higher than that of the untreated mangoes. Additionally, the firmness of the mangoes treated with 1-MCP + MT was considerably greater (*p* < 0.05) than that of the control mangoes during days 2–8 of storage.

### 3.2. Total Soluble Solids, Titratable Acidity, and Soluble Sugar Content

Total soluble solids content, titratable acidity and soluble sugar content are essential components of mango fruit that affect its taste, nutrition, and storage quality. As shown in Figure 2A and Appendix A, after an initial increase, the total soluble solids content of untreated mango fruit decreased and then slightly increased again. The total soluble solids content in 1-MCP and 1-MCP + MT-treated mango fruit increased continuously during storage and was significantly higher than that of untreated mangoes from storage days 8 to 10 (*p* < 0.05). However, there was no significant difference between the total soluble solids level of the 1-MCP and the 1-MCP + MT-treated mango fruits.

Figure 2B and Appendix A displays an overall downward trend in the titratable acidity content of all samples during storage. All treatment groups experienced a gradual decline in titratable acidity content from days 0–4 before a sharp decline. Within the period of storage days 4–6, the titratable acidity content of the 1-MCP-treated mangoes remained at a notably higher level than the control. However, the titratable acidity content reduction was remarkably delayed by 1-MCP + MT treatment during storage. Interestingly, the 1-MCP + MT-treated mangoes demonstrated a significantly higher level of titratable acidity than the other two groups (*p* < 0.05).

The soluble sugar level of mango pulp among the three groups exhibited an increasing tendency during 0–2 days and 4–8 days but decreased during storage days 2–4 and days 8–10 (Figure 2C, Appendix A). During storage, the 1-MCP + MT treatment produced more soluble sugar than the other two groups. Further analysis showed that within the 6–10-day storage period, the soluble sugar content of the 1-MCP + MT group had a significantly higher (*p* < 0.05) soluble sugar level than the 1-MCP and control groups. Furthermore, the 1-MCP treatment only maintained a significantly elevated content of soluble sugar during the early and late periods compared to the control (*p* < 0.05).

### 3.3. Ascorbic Acid and Total Phenolic Content

Ascorbic acid serves as one of the most essential nutritional factors to estimate the quality of fruits and vegetables. Moreover, it functions effectively as an antioxidant in plants. The concentrations of ascorbic acid had a divergent downward trend throughout the storage process in all three groups, as shown in Figure 3A and Appendix A. The ascorbic acid content decreased sharply in the control mangoes during days 2–4, resulting in a lower content of ascorbic acid than in the two treatment groups during days 4–10. Statistical analysis revealed that the 1-MCP group maintained a significantly higher content of ascorbic acid than the control group on day 4 and during days 0–3 of storage (*p* < 0.05). Within days 4–10 of storage, the 1-MCP + MT group retained the highest ascorbic acid content.

The total phenolic content for the three groups during storage is shown in Figure 3B and Appendix A. After 4 days of storage, the total phenolic content of mangoes without treatment increased rapidly to 407.26 mg GAE/kg and then declined quickly. However, the total phenolic content in the two treatment groups raised comparatively slowly during the storage period, except for a slight decrease in storage days 0–6. It was also shown that the total phenolic content of the 1-MCP + MT and 1-MCP-treated mangoes was significantly lower than that of the control (*p* < 0.05) on days 2–6. Moreover, the total phenolic content of mangoes with 1-MCP + MT treatment differed significantly from that of the 1-MCP-treated mangoes at days 4–10 (*p* < 0.05).

### 3.4. O_2_^−.^ Generation Rate and Malondialdehyde Content 

O_2_^−.^ as an intracellular ROS can cause irreversible damage to cells when excessively accumulated. Excess intracellular ROS such as O_2_^−.^ may harm cells irreversibly. Figure 4A and Appendix A indicated that 1-MCP + MT treatment inhibited O_2_^−.^ generation during storage (*p* < 0.05). The O_2_^−.^ generation rate in 1-MCP-treated mango was dramatically different from that in control mango without 1-MCP on day 6. Nevertheless, the 1-MCP + MT treatment was more effective in decreasing O_2_^−.^ accumulation.

The malondialdehyde content increased continuously over time but most notably during storage days 2–8 (Figure 4B, Appendix A). The malondialdehyde level of the control mango was 22.28 nmol/100 g on day 6, which was 213% and 202% higher than that of the 1-MCP + MT and 1-MCP-treated mango, respectively. These results revealed that the 1-MCP + MT-treated mangoes maintained the lowest malondialdehyde content.

### 3.5. Superoxide Dismutase, Catalase, Ascorbate Peroxidase, and Peroxidase Activities

Relatively higher superoxide dismutase activity was recorded in mango pulp with the 1-MCP and 1-MCP + MT treatments (Figure 5A, Appendix A), and the 1-MCP + MT treatment was more prominent in stimulating superoxide dismutase activity. The superoxide dismutase activity of 1-MCP + MT-treated mango pulp was considerably higher at storage days 2–6 and 10 compared to mangoes in the 1-MCP treatment group and the control group (*p* < 0.05). Compared to the control group, a significantly higher superoxide dismutase activity in mango with 1-MCP treatment was recorded on storage days 4–6 and on storage day 10 (*p* < 0.05).

Throughout the storage period, the activity of catalase in the control mangoes was significantly lower than in mangoes with 1-MCP treatment. The catalase activity in 1-MCP-treated mangoes had a significantly higher value (*p* < 0.05) than that of control mango at storage days 2–10. Additionally, the catalase activity in 1-MCP + MT-treated mangoes exhibited lower values on storage days 2–6 and higher values on day 8 than mangoes with 1-MCP treatment (Figure 5B, Appendix A). In the early (2 days) and late (8–10 days) of storage, 1-MCP + MT treatment significantly increased catalase activity compared to the control (*p* < 0.05).

The ascorbate peroxidase activity of the control mangoes consistently declined from 0 to 8 days and then increased (Figure 5C, Appendix A). On storage day 8, the lowest activity of ascorbate peroxidase was achieved in the control group, which was 51.1% and 48.0% of the 1-MCP and 1-MCP + MT groups, respectively. After 4–8 days of storage, the ascorbate peroxidase activity in the control mangoes was significantly lower than that of 1-MCP + MT-treated mango (*p* < 0.05). In addition, on the 6th day, the ascorbate peroxidase activity in 1-MCP + MT-treated mangoes was also significantly higher than that of the 1-MCP-treated mangoes (*p* < 0.05). Figure 5C shows that, in comparison with the control, the 1-MCP treatment had an impact on maintaining the ascorbate peroxidase activity at higher levels for days 4–10, and a significant difference was recorded (*p* < 0.05).

1-MCP and 1-MCP + MT treatments induced noticeably higher peroxidase activities (*p* < 0.05) during the mango storage period compared to the control group (Figure 5D, Appendix A). During the storage period, the peroxidase activity in the control group was relatively stable. In the 1-MCP group, the peroxidase activity escalated at storage days 0–2, decreased during storage days 2–4, and then remained relatively stable from days 4 to 10. The 1-MCP + MT group showed two periods (days 0–4 and days 6–8) of increasing peroxidase activity. Further investigation demonstrated that the mangoes with 1-MCP + MT treatment had significantly higher peroxidase activity than the 1-MCP-treated mangoes on days 4–8 (*p* < 0.05).

## 4. Discussion

Exogenous MT exhibited a positive effect on regulating the ripeness of postharvest fruit while simultaneously having a negative effect on fruit senescence [21]. Previous studies have reported that MT accelerates the ripening of tomatoes [38] and grape berries [28] by promoting ethylene biosynthesis. However, contrary conclusions were drawn from the studies on bananas [39], strawberries [40], pears [41], and mangoes [42] in which the use of MT inhibited the production of ethylene and delayed the senescence and ripening of fruits. Fruit senescence is linked to a decrease in quality and an imbalance in active oxygen metabolism. In this study, throughout the storage at 25 °C, mangoes treated with 1-MCP + MT demonstrated greater antioxidant activity and quality than those treated with 1-MCP alone. 

The mango in the 1-MCP + MT group maintained a stable *a** value and inhibited the increase in *b** during storage, which reduced the colour change in the mango epidermis and maintained good appearance quality. In particular, the 1-MCP + MT treatment only caused the smallest change in *a** values of the mango’s epidermis, which meant that these mangoes remained green after 10 days of storage. The mangoes were significantly yellow in the 1-MCP and control groups. This result is inconsistent with the result reported by Rastegar et al. [43] in which the single use of MT showed no impact on the colour change in mangoes. The synergy of 1-MCP and MT could maintain mango firmness and produce greater titratable acidity and soluble sugar content than 1-MCP alone or in the control. The total soluble solids content of mangoes was reduced during days 2–6 of storage due to the addition of 1-MCP alone and 1-MCP + MT. The reason for this might be that the control mangoes matured faster after 6 days of storage. In other studies, employing MT alone had no effect on mango colour change, titratable acidity, or total soluble solids content [43]. Studies have shown that strawberries treated with MT alone have a lower ascorbic acid content [44], whereas, in this study, the 1-MCP + MT treatment effectively delayed the decrease in the ascorbic acid content of mangoes and had a higher content than that of the 1-MCP group. Therefore, in terms of obtaining higher-quality mangoes, the combination of 1-MCP and MT is superior to 1-MCP alone.

Exogenous MT has been proven in several studies to increase phenolic synthesis [27,45,46]. Nevertheless, in our investigation, the total phenolic content of the control mangoes significantly changed after storage but changed less in the 1-MCP and 1-MCP + MT-treated mangoes. Plant phenolic compounds assist in disease resistance via antibacterial and bactericidal activities, free radical removal, and anti-lipid peroxidation [47]. In the late period of storage, the reduction in total phenolic levels in the control mangoes reduced the ability of the fruit to resist pathogens, thereby accelerating decay (Figure 1E). At low concentrations, ROS act as signal molecules in plants and aid in defence responses; however, excessive amounts can lead to oxidative damage and enhance the ripening and senescence processes. Antioxidant enzymes and antioxidants in plants affect ROS removal [33,37]. In this study, we discovered that the combination of 1-MCP and MT played a primary role in promoting ascorbic acid and strengthening superoxide dismutase, ascorbate peroxidase, and peroxidase activities. In addition, their combination also inhibited the accumulation of O_2_^−.^ and malondialdehyde while increasing catalase activity in the latter storage period (Figure 5), which helped to decrease ROS damage to fruit tissue and delayed the process of senescence and ripening. Different storage stages were associated with the peak activities of different antioxidant enzymes, indicating that each of them played a role in scavenging ROS at different stages. The antioxidative effects of 1-MCP, which reduces ROS production in other fruits [13,14,15], are enhanced by MT.

## 5. Conclusions

The effects of the combined treatment of 1-MCP and MT improved the visual and quality characteristics of mangoes during storage, maintained higher ascorbic acid content, decreased O_2_^−.^ generation rates, reduced the malondialdehyde content, and maintained higher antioxidant enzyme activity levels. Whether in terms of quality or the regulation of the active metabolism of mangoes, the effects of the combination of 1-MCP and MT were superior to those of 1-MCP alone. Hence, the 1-MCP and MT combined treatment may be a potential alternative in the postharvest technology of mango fruits during storage.

## Figures and Tables

**Figure 1 foods-12-01979-f001:**
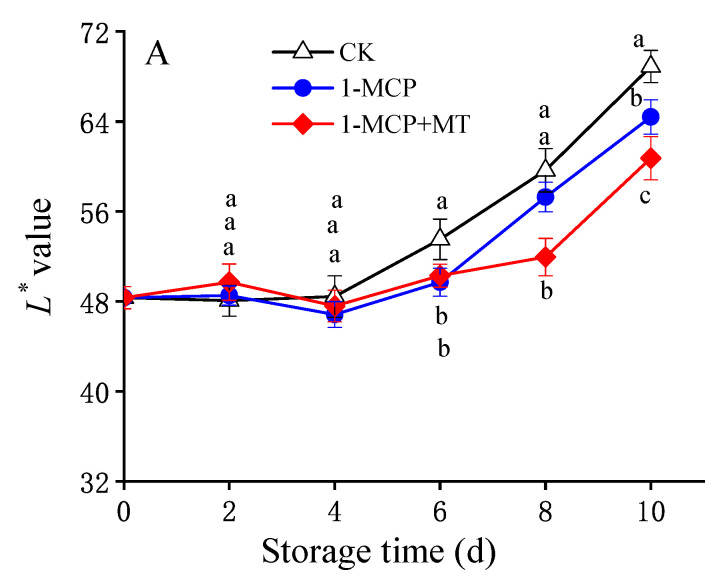
The effects of the 1-MCP and 1-MCP + MT treatments on *L** (**A**), *a** (**B**), and *b** (**C**) values of mango peel firmness (**D**) and visual appearance (**E**) of mango fruit during storage at 25 °C. Error bars indicate the standard deviation (SD) of six (*L**, *a**, and *b** value) or eight (firmness) replicates. Significant differences (*p* < 0.05) among different treatments on the same storage day are indicated by different lowercase letters.

**Figure 2 foods-12-01979-f002:**
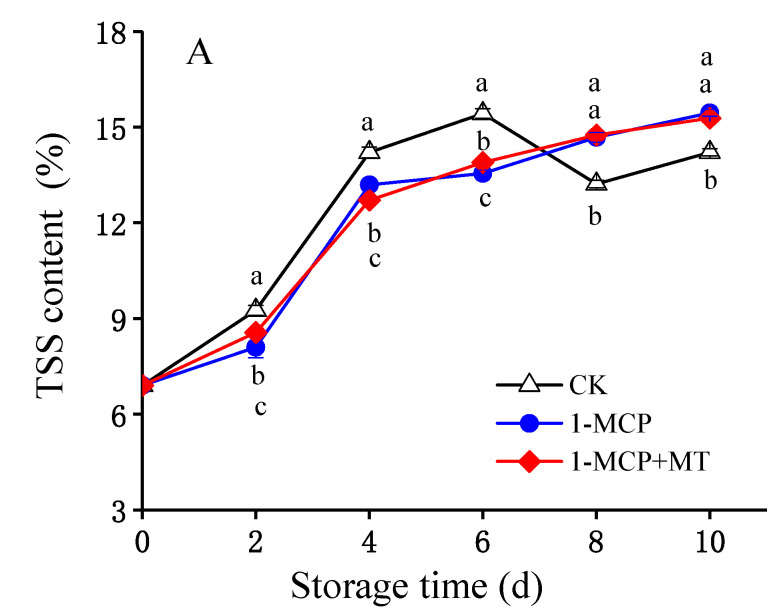
Effects of 1-MCP and 1-MCP + MT treatment on total soluble solids (TSS) content (**A**), titratable acidity (TA) (**B**), and soluble sugar (SS) content (**C**) in mango fruit during storage at 25 °C. Error bars indicate the SD of three replicates. Significant differences (*p* < 0.05) among different treatments on the same storage day are indicated by different lowercase letters.

**Figure 3 foods-12-01979-f003:**
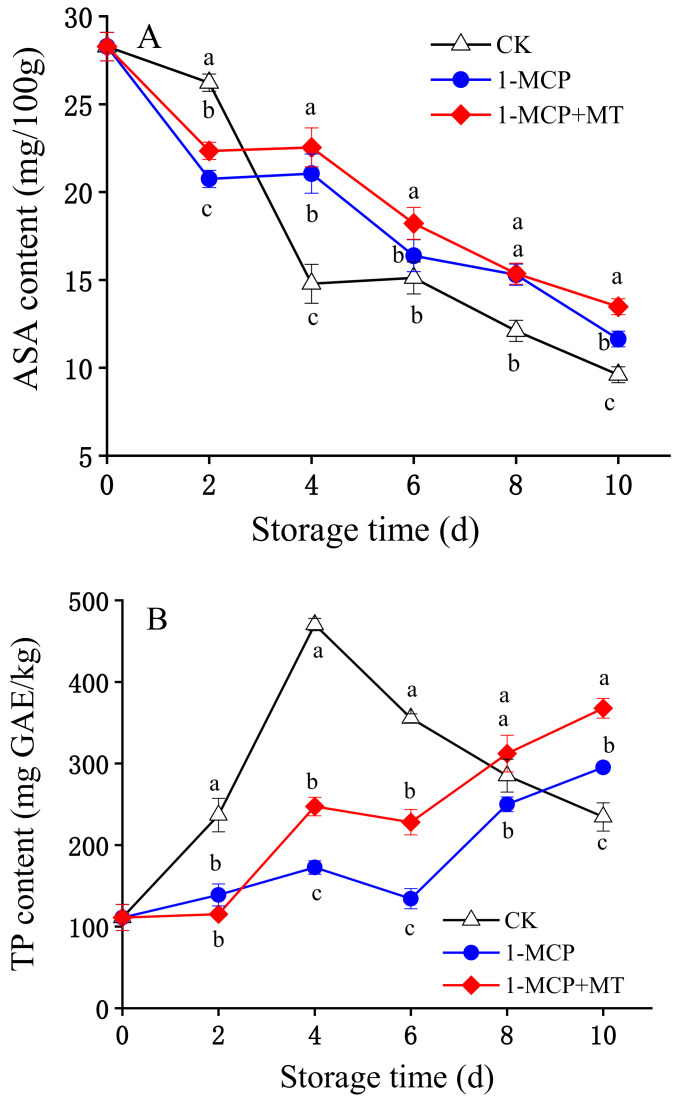
The effects of 1-MCP and 1-MCP + MT treatment on the ascorbic acid (ASA) content (**A**) and total phenolic content (TP) (**B**) of mango fruit during storage at 25 °C. Error bars indicate the SD of three replicates. Significant differences (*p* < 0.05) among different treatments on the same storage day are indicated by different lowercase letters.

**Figure 4 foods-12-01979-f004:**
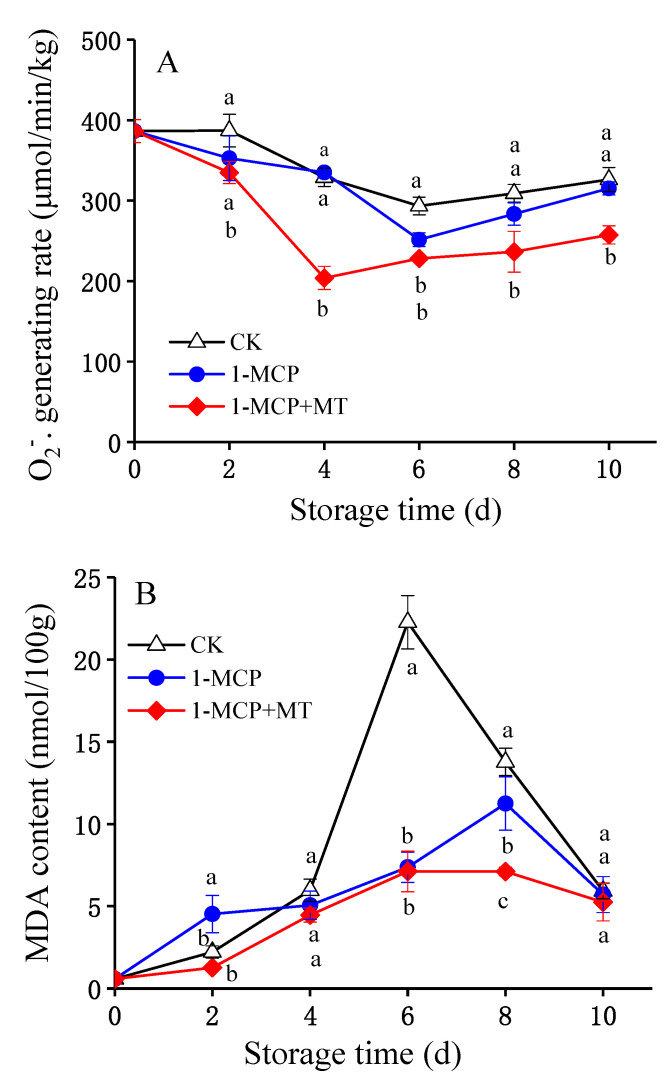
The effects of 1-MCP and 1-MCP + MT treatment on O_2_^-.^ generating rate (**A**) and malondialdehyde (MDA) content (**B**) in mangoes during storage at 25 °C. Error bars indicate the SD of three replicates. Significant differences (*p* < 0.05) among different treatments on the same storage day are indicated by different lowercase letters.

**Figure 5 foods-12-01979-f005:**
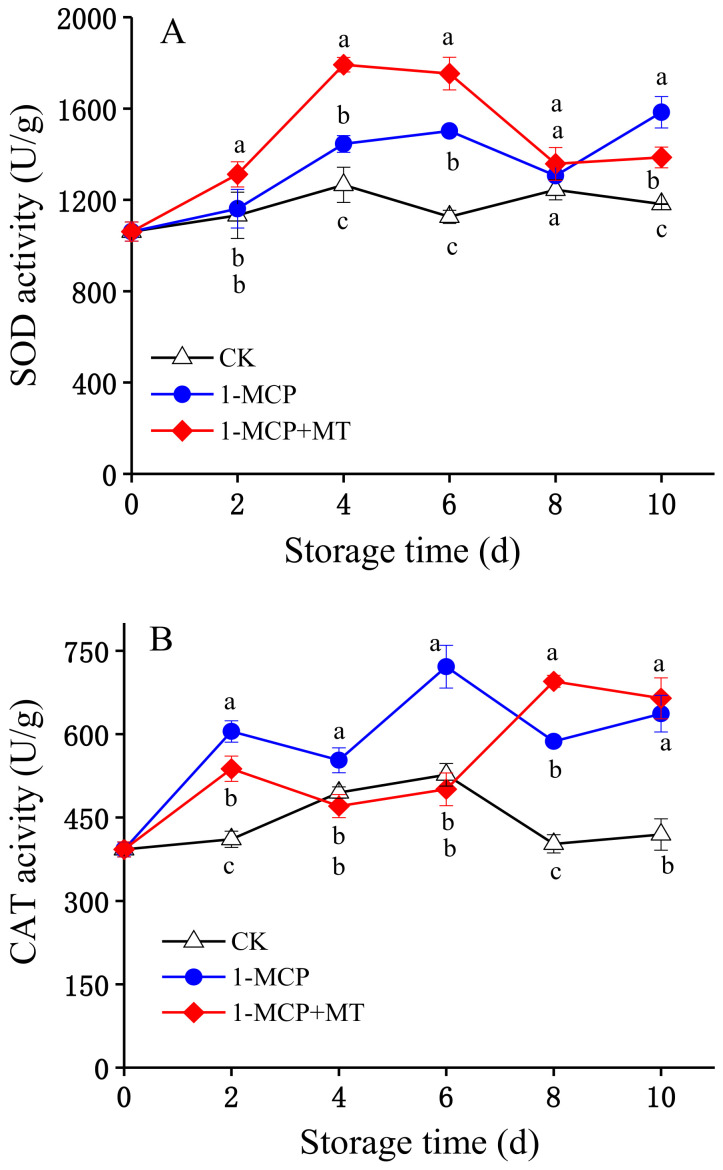
The effects of 1-MCP and 1-MCP + MT treatment on superoxide dismutase (SOD) (**A**), catalase (CAT) (**B**), ascorbate peroxidase (APX) (**C**), and peroxidase (POD) (**D**) activities in mangoes during storage at 25 °C. Error bars indicate the SD of three replicates. Significant differences (*p* < 0.05) among different treatments on the same storage day are indicated by different lowercase letters.

## Data Availability

The data are available from the corresponding author.

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
