# Peer review of "The Effects of Combined 1-Methylcyclopropene and Melatonin Treatment on the Quality Characteristics and Active Oxygen Metabolism of Mango Fruit during Storage"

_foods, 2023, doi:10.3390/foods12101979_

Round 1
Reviewer 1 Report
The aim of the study was to apply 1-methylcyclopropene and melatonin to mango fruits for quality maintaining during storage.
Although the title starts with 'Synergistic effect of...`, this aspect is not further discussed within the paper. What is the mechanism of synergy between the two agents? Which are the improvements brought by the addition of the second agent?
In the graphs values for CK appears, but this group of letter is not previously defined. Is is the control sample?
The conclusions of the study must be presented more clearly; for example was the aim of the study fulfilled? What are the authors future perspectives after conducting this study?
Also, English language could be improved, the along the article (for example in the abstract: `or with a combination of...`, `under the following conditions, etc.)
Reviewer 2 Report
I have reviewed the manuscript entitled: “Synergistic effect of 1-methylcyclopropene and melatonin on the quality characteristics and active oxygen metabolism of mango fruit during storage”. The manuscript evaluates the effect of the addition of 1-methylcyclopropene and melatonin on some the quality characteristics (soluble sugar, ascorbic acid, firmness, colour, titratable acidity, ascorbic acid, and total phenolic content) and active oxygen metabolism (malondialdehyde content, superoxide anion generation rate, and CAT, SOD, POD, and APX activities) of post-harvest mango. The manuscript shows that the antioxidant effects of 1-methylcyclopropene are enhanced by melatonin, having positive effects on mango senescence and quality.
The abstract is well written and summarizes the main effects of 1-methylcyclopropene alone and both 1-methylcyclopropene and melatonin on the evaluated parameters.
· The expression “relatively stable” should be avoided. Use only the words “significant” (p < 0.05) or “non-significant” (p > 0.05).
Introduction is well addressed and clear. It includes a general review on the impact of oxygen metabolism on fruit senescence and presents the main results from previous studies about the role of 1-MCP and melatonin in preserving fruit quality. However, some sentences should be reviewed, for instance:
· “The objective of this work was to assess the synergistic effect of 1-MCP and MT on the firmness, nutrients, and antioxidant activity of fresh mango fruit during storage.” à As stated in the Abstract “Compared to 1-MCP treatment alone, 1-MCP + MT treated mango exhibited higher quality and stronger regulation of the active metabolism during the storage”. However, in order to have a synergistic effect, the effect of 1-MCP + MT should be greater than the sum of the effects of each compound. Thus, it is not possible to conclude that 1-MCP + MT have a synergistic effect. The effect of MT alone should be studied and compared.
· “Melatonin (N-acetyl-5-methoxytryptamine, MT) is composed of serotonin (metabolised by tryptophan [19,20])” à please rewrite this sentence. Melatonin is derived from serotonin, resulting from the metabolism of tryptophan.
Materials and methods are well summarized and have been previously described in other publications.
· The standard for TP level determination was gallic acid (GA) equivalent. The level of TP was presented as mg GA·kg−1. à if the standard used was gallic acid, please remove “equivalent”. In this case, the results should be expressed as GAE (gallic acid equivalents).
Results are somehow difficult to read and are presented in a very exhaustive manner. They should mostly emphasise the significant differences. Although the Figures 1 to 4 show the general results in a very straightforward manner, the values of each evaluated parameter, at each storage time, should be presented in a Supplementary Table. The average and standard deviation should be included.
· CK (figures): please add the legend of CK or change it for “no treatment”, for instance.
· I suggest changing the expression “remarkable differences” by “significant differences” and presenting the p-value.
· The results show several acronyms that make the text difficult to read. I suggest avoiding the use of some acronyms, as ASA, TSS, SS, TA. The words “ascorbic acid”, “total soluble solids”, etc, can be used instead.
The discussion is clear, and I believe it can be improved if less acronyms are used.
The conclusions are just the presentation of the main results. They should also emphasize the main ideas of the manuscript and the relevance of the work.
Round 2
Reviewer 2 Report
I have reviewed the modified version of the manuscript entitled: “Effects of combined treatment of 1-methylcyclopropene and melatonin on the quality characteristics and active oxygen metabolism of mango fruit during storage”. Authors have improved most of the topics previously suggested. However, I believe the results must be improved and some editing for English language is required throughout the manuscript.
Other minor suggestions:
· Page 1: “The mango fruit was then stored for 10 days under the following conditions: 85%–90% relative humidity and 25 ℃.” à The mango fruit was then stored for 10 days at 25 °C and at 85%–90% relative humidity.
· Page 1: “Nevertheless, maintained higher total phenol content only at the later stage of storage” à English must be improved.
· Page 2: “Melatonin (N-acetyl-5-methoxytryptamine, MT) is synthesized from serotonin, which is metabolized from tryptophan [19,20]), which is commonly…” à The repetition of the word “which” should be avoided.
· Page 2: “and reduce the production of ROS 27.” à and reduce the production of ROS [27].
· Page 3: “Measurement of total soluble solids and titratable acidity and total soluble sugar contents” à Measurement of total soluble solids, total soluble sugar content and titratable acidity
· Page 4: “Quantification of ascorbic acid and total phenol” à Quantification of ascorbic acid and total phenolic content. Replace the word “phenol” by “phenolic” throughout the manuscript.
· Page 7: SS level à soluble sugar level.
· Page 9: “1-MCP + MT group retained a markedly higher ascorbic acid content than the other two groups” à 1-MCP + MT group retained the highest ascorbic acid content.
· Page 9: Please check the letters “a, b, and c” of days 6 and 10.
· Page 10: Malondialdehyde content increased continuously with increasing time. à English must be improved.
· Page 10: “These results revealed that 1-MCP + MT had a more significant effect (P < 0.05) on retarding the accumulation of malondialdehyde.” à English must be improved.
· Figures: Please write the meaning of all the acronyms (MDA, PC, etc) in the legend of each figure.
